# Individual Privacy Accounting with Gaussian Differential Privacy

**Antti Koskela**[1,2]**, Marlon Tobaben**[2] **and Antti Honkela**[2]

[1] Nokia Bell Labs, Espoo, Finland
[2] Helsinki Institute for Information Technology HIIT,
Department of Computer Science, University of Helsinki, Finland

## Abstract

Individual privacy accounting enables bounding differential privacy (DP) loss individually for each participant involved in the analysis. This can be informative as often the individual privacy losses are considerably smaller than those indicated by the DP bounds that are based on considering worst-case bounds at each data access. In order to account for the individual privacy losses in a principled manner, we need a privacy accountant for adaptive compositions of randomised mechanisms, where the loss incurred at a given data access is allowed to be smaller than the worst-case loss. This kind of analysis has been carried out for the Rényi differential privacy by Feldman and Zrnic (2021), however not yet for the so-called optimal privacy accountants. We make first steps in this direction by providing a careful analysis using the Gaussian differential privacy which gives optimal bounds for the Gaussian mechanism, one of the most versatile DP mechanisms.

## 1 Introduction

Differential privacy (DP) Dwork et al. (2006) provides means to accurately bound the compound privacy loss of multiple accesses to a database. Common differential privacy composition accounting techniques such as Rényi differential privacy (RDP) based techniques (Mironov, 2017; Wang et al., 2019; Zhu and Wang, 2019; Mironov et al., 2019) or so called optimal accounting techniques (Koskela et al., 2020; Gopi et al., 2021; Zhu et al., 2022) require that the privacy parameters of all algorithms are fixed beforehand. Rogers et al. (2016) were the first to analyse fully adaptive compositions, wherein the mechanisms are allowed to be selected adaptively. Rogers et al. (2016) introduced two objects for measuring privacy in fully adaptive compositions: privacy filters, which halt the algorithms when a given budget is exceeded, and privacy odometers, which output bounds on the privacy loss incurred so far. Whitehouse et al. (2022) have tightened these composition bounds using filters to match the tightness of the so called advanced composition theorem (Dwork and Roth, 2014). Feldman and Zrnic (2021) obtain similar $(\varepsilon, \delta)$-asymptotics via RDP analysis.

In their analysis using the Rényi differential privacy, Feldman and Zrnic (2021) consider individual filters, where the algorithm stops releasing information about the data elements that have exceeded a pre-defined RDP budget. This kind of individual analysis has not yet been considered for the optimal privacy accountants. We make first steps in this direction by providing a fully adaptive individual DP analysis using the Gaussian differential privacy (Dong et al., 2022). Our analysis leads to tight bounds for the Gaussian mechanism and it is based on determining a certain supermartingale for the hockey-stick divergence and on using similar proof techniques as in the RDP-based fully adaptive composition results of Feldman and Zrnic (2021). We also consider measuring the individual $(\varepsilon, \delta)$-privacy losses using the so-called privacy loss distributions (PLDs). Using the Blackwell theorem, we can in this case rely on the results of (Feldman and Zrnic, 2021) to construct an approximative $(\varepsilon, \delta)$-

2022 Trustworthy and Socially Responsible Machine Learning (TSRML 2022) co-located with NeurIPS 2022.

accountant that often leads to smaller individual $\varepsilon$-values than commonly used RDP accountants. We show that individual $\varepsilon$-values can be accurately approximated in $\mathcal{O}(n)$-time, where $n$ is the number of discretisation points for the PLDs. This is described in Appendix E.

As an observation of independent interest, we experimentally notice that individual filtering leads to a disparate loss of accuracies among subgroups when training a neural network using DP gradient descent.

## 2 Background

We first shortly review the required definitions and results for our analysis. For more detailed discussion, see e.g. (Dong et al., 2022) and (Zhu et al., 2022).

An input data set containing $N$ data points is denoted as $X = (x_1, \ldots, x_N) \in \mathcal{X}^N$, where $x_i \in \mathcal{X}$, $1 \le i \le N$. We say $X$ and $X'$ are neighbours if we get one by adding or removing one element in the other (denoted $X \sim X'$). To this end, similarly to Feldman and Zrnic (2021), we also denote $X^{-i}$ the dataset obtained by removing element $x_i$ from $X$, i.e.

$$X^{-i} = (x_1, \ldots, x_{i-1}, x_{i+1}, \ldots, x_N).$$

A mechanism $\mathcal{M}$ is $(\varepsilon, \delta)$-DP if its outputs are $(\varepsilon, \delta)$-indistinguishable for neighbouring datasets.

**Definition 1.** *Let $\varepsilon \ge 0$ and $\delta \in [0, 1]$. Mechanism $\mathcal{M} : \mathcal{X}^n \to \mathcal{O}$ is $(\varepsilon, \delta)$-DP if for every pair of neighbouring datasets $X, X'$, every measurable set $E \subset \mathcal{O}$,*

$$\mathbb{P}(\mathcal{M}(X) \in E) \le \mathrm{e}^\varepsilon \mathbb{P}(\mathcal{M}(X') \in E) + \delta.$$

*We call $\mathcal{M}$ tightly $(\varepsilon, \delta)$-DP, if there does not exist $\delta' < \delta$ such that $\mathcal{M}$ is $(\varepsilon, \delta')$-DP.*

The $(\varepsilon, \delta)$-DP bounds can also be characterised using the Hockey-stick divergence. For $\alpha > 0$ the hockey-stick divergence $H_\alpha$ from a distribution $P$ to a distribution $Q$ is defined as

$$H_\alpha(P||Q) = \int [P(t) - \alpha \cdot Q(t)]_+ \ \mathrm{d}t,$$

where for $x \in \mathbb{R}$, $[x]_+ = \max\{0, x\}$. Tight $(\varepsilon, \delta)$-values for a given mechanism can be obtained using the hockey-stick-divergence:

**Lemma 2** (Zhu et al. 2022). *For a given $\varepsilon \ge 0$, tight $\delta(\varepsilon)$ is given by the expression*

$$\delta(\varepsilon) = \max_{X \sim X'} H_{\mathrm{e}^\varepsilon}(\mathcal{M}(X)||\mathcal{M}(X')).$$

Thus, if we can bound the divergence $H_{\mathrm{e}^\varepsilon}(\mathcal{M}(X)||\mathcal{M}(X'))$ accurately, we also obtain accurate $\delta(\varepsilon)$-bounds. To this end we consider so called dominating pairs of distributions:

**Definition 3** (Zhu et al. 2022). *A pair of distributions $(P, Q)$ is a dominating pair of distributions for mechanism $\mathcal{M}(X)$ if for all neighbouring datasets $X$ and $X'$ and for all $\alpha > 0$,*

$$H_\alpha(\mathcal{M}(X)||\mathcal{M}(X')) \le H_\alpha(P||Q).$$

*If the equality holds for all $\alpha$ for some $X, X'$, then $(P, Q)$ is tightly dominating.*

Dominating pairs of distributions also gives upper bounds for adaptive compositions:

**Theorem 4** (Zhu et al. 2022). *If $(P, Q)$ dominates $\mathcal{M}$ and $(P', Q')$ dominates $\mathcal{M}'$, then $(P \times P', Q \times Q')$ dominates the adaptive composition $\mathcal{M} \circ \mathcal{M}'$.*

To convert the hockey-stick divergence from $P \times P'$ to $Q \times Q'$ into an efficiently computable form, we consider so called privacy loss random variables.

**Definition 5.** *Let $P$ and $Q$ be probability density functions. We define the privacy loss function $\mathcal{L}_{P/Q}$ as*

$$\mathcal{L}_{P/Q}(t) = \log \frac{P(t)}{Q(t)}.$$

*We define the privacy loss random variable (PRV) $\omega_{P/Q}$ as*

$$\omega_{P/Q} = \mathcal{L}_{P/Q}(t), \quad t \sim P(t).$$

With slight abuse of notation, we denote the probability density function of the random variable $\omega_{P/Q}$ by $\omega_{P/Q}(t)$, and call it the privacy loss distribution (PLD).

**Theorem 6** (Gopi et al. 2021). *The $\delta(\varepsilon)$-bounds can be represented using the following representation that involves the PRV:*

$$H_{e^\varepsilon}(P||Q) = \mathop{\mathbb{E}}_{t \sim P}\left[1 - e^{\varepsilon - \mathcal{L}_{P/Q}(t)}\right]_+ = \mathop{\mathbb{E}}_{s \sim \omega_{P/Q}}\left[1 - e^{\varepsilon - s}\right]_+. \tag{2.1}$$

*Moreover, if $\omega_{P/Q}$ is the PRV for the pair of distributions $(P, Q)$ and $\omega_{P'/Q'}$ the PRV for the pair of distributions $(P', Q')$, then the PRV for the pair of distributions $(P \times P', Q \times Q')$ is given by $\omega_{P/Q} + \omega_{P'/Q'}$.*

When we set $\alpha = e^\varepsilon$, the following characterisation follows from Lemma 2 and Theorem 6.

**Corollary 7.** *If the pair of distributions $(P, Q)$ is a dominating pair of distributions for a mechanism $\mathcal{M}$, then for all neighbouring datasets $X$ and $X'$ and for all $\varepsilon \in \mathbb{R}$,*

$$\mathop{\mathbb{E}}_{t \sim \mathcal{M}(X)}\left[1 - e^{\varepsilon - \mathcal{L}_{\mathcal{M}(X)/\mathcal{M}(X')}(t)}\right]_+ \leq \mathop{\mathbb{E}}_{t \sim P}\left[1 - e^{\varepsilon - \mathcal{L}_{P/Q}(t)}\right]_+.$$

We will in particular consider the Gaussian mechanism and its subsampled variant.

**Example: hockey-stick divergence between two Gaussians.** Let $x_0, x_1 \in \mathbb{R}$, $\sigma \geq 0$, and let $P$ be the density function of $\mathcal{N}(x_0, \sigma^2)$ and $Q$ the density function of $\mathcal{N}(x_1, \sigma^2)$. Then, the PRV $\omega_{P/Q}$ is distributed as (Lemma 11 by Sommer et al., 2019)

$$\omega_{P/Q} \sim \mathcal{N}\left(\frac{(x_0 - x_1)^2}{2\sigma^2}, \frac{(x_0 - x_1)^2}{\sigma^2}\right). \tag{2.2}$$

Thus, in particular: $H_\alpha(P||Q) = H_\alpha(Q||P)$ for all $\alpha > 0$. Plugging in PLD $\omega_{P/Q}$ to the expression (2.1), we find that for all $\varepsilon \geq 0$, the hockey-stick $H_{e^\varepsilon}(P||Q)$ is given by the expression

$$\delta(\varepsilon) = \Phi\left(-\frac{\varepsilon\sigma}{\Delta} + \frac{\Delta}{2\sigma}\right) - e^\varepsilon \Phi\left(-\frac{\varepsilon\sigma}{\Delta} - \frac{\Delta}{2\sigma}\right), \tag{2.3}$$

where $\Phi$ denotes the CDF of the standard univariate Gaussian distribution and $\Delta = |x_0 - x_1|$. This formula was given already by Balle and Wang (2018).

If $\mathcal{M}$ is of the form $\mathcal{M}(X) = f(X) + Z$, where $f : \mathcal{X}^N \to \mathbb{R}^d$ and $Z \sim \mathcal{N}(0, \sigma^2 I_d)$, and $\Delta = \max_{X \simeq X'} \|f(X) - f(X')\|_2$, then for $x_0 = 0$, $x_1 = \Delta$, $(P, Q)$ of the above form gives a tightly dominating pair of distributions for $\mathcal{M}$ (Zhu et al., 2022). Subsequently, by Theorem 6, $\mathcal{M}$ is $(\varepsilon, \delta)$-DP for $\delta(\varepsilon)$ given by (2.3).

Lemma 8 allows tight analysis of the subsampled Gaussian mechanism using the hockey-stick divergence. We state the result for the case of Poisson subsampling with sampling rate $\gamma$.

**Lemma 8** (Zhu et al. 2022). *If $(P, Q)$ dominates a mechanism $\mathcal{M}$ for add neighbors then $(P, (1 - \gamma) \cdot P + \gamma \cdot Q)$ dominates the mechanism $\mathcal{M} \circ S_{Poisson}$ for add neighbors and $((1 - \gamma) \cdot Q + \gamma \cdot P), P)$ dominates $\mathcal{M} \circ S_{Poisson}$ for removal neighbors.*

## 3 Fully Adaptive Compositions

In order to compute tight $\delta(\varepsilon)$-bounds for fully adaptive compositions, we determine a suitable random variable that gives us the bounds for fully adaptive compositions.

Our approach is motivated by the existing analysis for fully adaptive compositions by Feldman and Zrnic (2021). Similarly to Feldman and Zrnic (2021), we denote the mechanism corresponding to the fully adaptive composition of first $n$ mechanisms as

$$\mathcal{M}^{(n)}(X) = \left(\mathcal{M}_1(X), \mathcal{M}_2(\mathcal{M}_1(X), X), \ldots, \mathcal{M}_n(\mathcal{M}_1(X), \ldots, \mathcal{M}_{n-1}(X), X)\right)$$

and the sequence of outcomes of $\mathcal{M}^{(n)}(X)$ as $a^{(n)} = (a_1, \ldots, a_n)$, For two datasets $X$ and $X'$, define $\mathcal{L}_{X/X'}^{(n)}$ as the total privacy loss

$$\mathcal{L}_{X/X'}^{(n)} = \log\left(\frac{\mathbb{P}(\mathcal{M}^{(n)}(X) = a^{(n)})}{\mathbb{P}(\mathcal{M}^{(n)}(X') = a^{(n)})}\right)$$

and, given $a^{(n-1)}$, we define $\mathcal{L}^n_{X/X'}$ as the privacy loss of the mechanism $\mathcal{M}_n$,

$$\mathcal{L}^n_{X/X'} = \log\left(\frac{\mathbb{P}(\mathcal{M}_n(a^{(n-1)}, X) = a_n)}{\mathbb{P}(\mathcal{M}_n(a^{(n-1)}, X') = a_n)}\right).$$

Using the Bayes rule it follows that $\mathcal{L}^{(n)}_{X/X'} = \mathcal{L}^{(n-1)}_{X/X'} + \mathcal{L}^n_{X/X'} = \sum_{m=1}^{n} \mathcal{L}^m_{X/X'}$.

### 3.1 Gaussian Differential Privacy

Informally speaking, a randomised mechanism $\mathcal{M}$ is $\mu$-GDP, $\mu \geq 0$, if for all neighbouring datasets the outcomes of $\mathcal{M}$ are not more distinguishable than two unit-variance Gaussians $\mu$ apart from each other (Dong et al., 2022). We formalise GDP using pairs of dominating distributions:

**Lemma 9.** *A mechanism $\mathcal{M}$ is $\mu$-GDP, if and only if for all neighbouring datasets $X, X'$ and for all $\alpha > 0$:*

$$H_\alpha(\mathcal{M}(X)\|\mathcal{M}(X')) \leq H_\alpha\big(\mathcal{N}(0,1)\|\mathcal{N}(\mu,1)\big). \tag{3.1}$$

*Proof.* By Cor. 2.13 of Dong et al. (2022), a mechanism is $\mu$-GDP if and only it is $(\varepsilon, \delta)$-DP for all $\varepsilon \geq 0$, where

$$\delta(\varepsilon) = \Phi\left(-\frac{\varepsilon}{\mu} + \frac{\mu}{2}\right) - e^\varepsilon \Phi\left(-\frac{\varepsilon}{\mu} - \frac{\mu}{2}\right).$$

By (2.3), this is equivalent to the fact that for all neighbouring datasets $X, X'$ and for all $\varepsilon \geq 0$:

$$H_{e^\varepsilon}(\mathcal{M}(X)\|\mathcal{M}(X')) \leq H_{e^\varepsilon}(\mathcal{N}(0,1)\|\mathcal{N}(\mu,1)).$$

By Lemma 31 of (Zhu et al., 2022), $H_\alpha\big(\mathcal{M}(X)\|\mathcal{M}(X')\big) \leq H_\alpha\big(P, Q\big)$ for all $\alpha > 1$ if and only if

$$H_\alpha\big(\mathcal{M}(X)\|\mathcal{M}(X')\big) \leq H_\alpha\big(Q, P\big)$$

for all $0 < \alpha \leq 1$. As $P$ and $Q$ are Gaussians, we see from the form of the privacy loss distribution (2.2) that $H_\alpha\big(Q, P\big) = H_\alpha\big(P, Q\big)$ and that (3.1) holds for all $\alpha > 0$. $\qquad\square$

### 3.2 GDP Analysis of Fully Adaptive Compositions

Analogously to individual RDP parameters (A.1), we define the conditional GDP parameters as

$$\mu_m = \inf\{\mu \geq 0 : \mathcal{M}_m(\cdot, a^{(m-1)}) \text{ is } \mu\text{–GDP}\}. \tag{3.2}$$

By Lemma 9 above, this means that for all neighbouring datasets $X, X'$ and for all $\alpha > 0$:

$$H_\alpha(\mathcal{M}_m(X, a^{(m-1)}), \mathcal{M}_m(X', a^{(m-1)})) \leq H_\alpha(\mathcal{N}(\mu_m, 1)\|\mathcal{N}(0, 1)).$$

**Example: Private GD.** Suppose each mechanism $\mathcal{M}_i$, $i \in [k]$, is of the form $\mathcal{M}_i(X, a) = \sum_{x \in X} f(x, a) + \mathcal{N}(0, \sigma^2)$. Since the hockey-stick divergence is scaling invariant, and since the sensitivity of the deterministic part of $\mathcal{M}_i(X, a)$ is $\max_{x \in X} \|f(x, a^{(m-1)})\|_2$, we have that $\mu_m = \max_{x \in X} \|f(x, a^{(m-1)})\|_2 / \sigma$.

We now give our main theorem. The proof is given in Appendix B.

**Theorem 10.** *Let $k$ denote the maximum number of compositions. Suppose that, almost surely,*

$$\sum_{m=1}^{k} \mu_m^2 \leq B^2.$$

*Then, $\mathcal{M}^{(k)}(X)$ is $B$-GDP.*

## 4 Individual GDP Filter

Similarly to (Feldman and Zrnic, 2021), we can determine an individual GDP privacy filter (Alg. 1) that keeps track of individual privacy losses and adaptively drops the data elements for which the cumulative privacy loss is about to cross the pre-determined budget. Using Theorem 10, we can show that the output of Alg. 1 is $B$-GDP. We give more details in Section C.1.

---

**Algorithm 1** Individual GDP Filter Algorithm

---

Input: Budget $B$, maximum number of compositions $k$, initial value $a_0$.
**for** $j = 1, \ldots, k$ **do**

For each $i \in [N]$, find parameter $\mu_j^{(i)} \geq 0$ such that for all $\alpha > 0$,

$$H_\alpha\big(\mathcal{M}_j(X, a^{(j-1)})||\mathcal{M}_j(X^{-i}), a^{(j-1)})\big) \leq H_\alpha\big(\mathcal{N}(\mu_j^{(i)}, 1)||\mathcal{N}(0, 1)\big).$$

Define the active set $S_j$: $\quad S_j = \{x_i : \mathcal{F}_B(\mu_1^{(i)}, \ldots, \mu_{j+1}^{(i)}) = \text{CONT}\}$.
For all $x_i$: set $\mu_{j+1}^{(i)} = \mu_{j+1}^{(i)} \mathbf{1}\{x_i \in S_j\}$.
Compute $a_j = \mathcal{M}_j(a^{(j-1)}, S_j)$.
**end for**
**return** $a^{(j)}$.

---

**Theorem 11.** *Denote by $\mathcal{M}$ the output of Algorithm 1. $\mathcal{M}$ is B-GDP under remove neighbourhood relation, meaning that for all datasets $X \in \mathcal{X}^N$, for all $i \in [N]$ and for all $\alpha > 0$:*

$$\max\{H_\alpha\big(\mathcal{M}(X)||\mathcal{M}(X^{-i}))\big), H_\alpha\big(\mathcal{M}(X^{-i})||\mathcal{M}(X))\big)\} \leq H_\alpha\big(\mathcal{N}(B, 1)||\mathcal{N}(0, 1)\big).$$

*Proof.* The proof goes the same way as the proof for (Thm. 4.3 Feldman and Zrnic, 2021) which holds for the RDP filter. Let $\mathcal{F}_t$ denote the natural filter $\sigma(a^{(t)})$. First, we define a GDP filter as

$$\mathcal{F}_B(\mu_1, \ldots, \mu_t) = \begin{cases} \text{HALT}, & \text{if } \sum_{i=1}^t \mu_i^2 > B^2, \\ \text{CONT}, & \text{else}. \end{cases}$$

We see that the random variable $T = \min\{\min\{t : \mathcal{F}_B(\mu_1, \ldots, \mu_{t+1}) = \text{HALT}\}, k\}$ is a stopping time since $\{T = t\} \in \mathcal{F}_t$ since $\mu_{t+1} \in \mathcal{F}_t$. From the optimal stopping theorem (Protter, 2004) and the supermartingale property of $M_n(\varepsilon)$ (as defined in (A.5)) it follows that $\mathbb{E}[M_T(\varepsilon)] \leq M_0(\varepsilon)$ for all $\varepsilon \in \mathbb{R}$. By the reasoning of the proof of Thm.10 we have that Alg. 1 is $B$-GDP. $\square$

**Example: Private GD.** Suppose each mechanism $\mathcal{M}_i$, $i \in [k]$, is of the form $\mathcal{M}_i(X, a) = \sum_{x \in X} f(x, a) + \mathcal{N}(0, \sigma^2)$. Then, we have that $\mu_j^{(i)} = \|f(x_i, a^{(j-1)})\|_2/\sigma$.

## 5 Approximative $(\varepsilon, \delta)$-Filter via Blackwell's Theorem

We next consider a filter that can use any individual dominating pairs of distributions, not just Gaussians. Our method is based on using a privacy of a $\mu$-GDP mechanism as an individual filter and on using the Blackwell theorem and the existing RDP results by (Feldman et al., 2022) to obtain rigorous individual filter. The approximative part is that of evaluating the individual $\mu$-parameters (see Fig. 1 for en example). Details of the method are given in D.

To illustrate the differences between the individual $\varepsilon$-values obtained with an RDP accountant and with the approximative PLD-based accountant, we consider DP-SGD training of a small feedforward network for MNIST classification. We choose randomly a subset of 1000 data elements and compute their individual $\varepsilon$-values (see Fig. 1). To compute the $\varepsilon$-values, we compare RDP accounting (as implemented in the TensorFlow Privacy library) and the aforementioned approach based on PLDs. We train for 50 epochs with batch size 300, noise parameter $\sigma = 2.0$ and clipping constant $C = 5.0$.

## 6 Experiments with MIMIC-III: Group-Wise $\varepsilon$-values

We consider the phenotype classification task from a MIMIC-III benchmark library (Harutyunyan et al., 2019) on the clinical database MIMIC-III (Johnson et al., 2016), freely-available from PhysioNet (Goldberger et al., 2000). The task is a multi-label classification and aims to predict which of 25 acute care conditions are present in a patient's MIMIC-III record. We have trained a multi-layer perceptron to maximise the macro-averaged AUC-ROC, the task's primary metric. We train the model using DP-GD combined with the Adam optimizer, and use the individual GDP filtering algorithm 1. See Appendix F for further details.

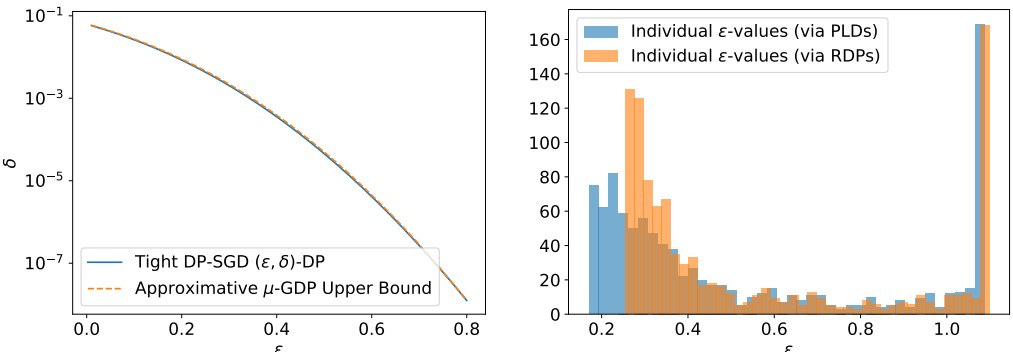

Figure 1: MNIST experiment. Left: Randomly chose data element and its accurate $(\varepsilon, \delta)$-curve after 50 epochs vs. the $\mu$-GDP upper bound approximation. Right: Comparison of individual $\varepsilon$-values obtained via RDPs and PLDs: histograms for randomly selected 1000 samples after 50 epochs ($\delta = 10^{-6}$). Computation using PLDs is better able to capture small individual $\varepsilon$-values.

To study the model behaviour between subgroups, we observe five non-overlapping groups of size 1000 from the test set and of size 400 from the train set by the present acute care condition: subgroup 0: no condition at all, subgroups 1 and 2: diagnosed with/not with Pneumonia, subgroups 3 and 4: diagnosed with/not with acute myocardial infarction (heart attack). Similarly as Yu et al. (2022), we see a correlation between individual $\varepsilon$-values and model accuracies across the subgroups: the groups with the best privacy protection (smallest average $\varepsilon$-values) have also the smallest average training and test losses. Fig. 2 shows that after running the filtered DP-GD beyond the worst-case $\varepsilon$-threshold for a number of iterations, both the training and test loss get smaller for the best performing group and larger for other groups. Similarly as DP-SGD has a disparate impact on model accuracies across subgroups (Bagdasaryan et al., 2019), we find that while the individual filtering leads to more equal group-wise $\varepsilon$-values, it leads to even larger differences in model accuracies. Here, one could alternatively consider other than algorithmic solutions for balancing the privacy protection among subgroups, by, e.g., collecting more data from the groups with the weakest privacy protection according to the individual $\varepsilon$-values (Yu et al., 2022).

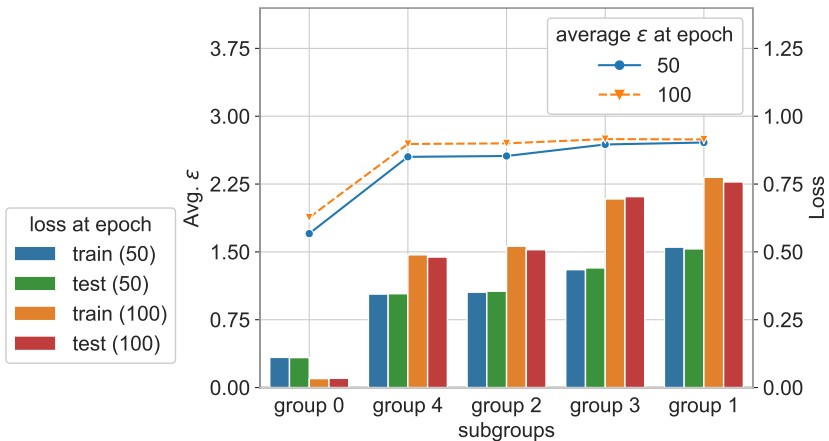

Figure 2: MIMIC III experiment and individual filtering for private GD. Comparing the test losses, training losses and average privacy losses before and after filtering has started (at 50 epochs). The filtering has a further disparate impact on model accuracies across subgroups.

**Acknowledgements**

The authors acknowledge CSC – IT Center for Science, Finland, and the Finnish Computing Competence Infrastructure (FCCI) for computational and data storage resources.

Our experiments use the MIMIC-III data set of pseudonymised health data by permission of the data providers. The data was processed according to usage rules defined by the data providers, and all reported results are anonymised. All the code related to MIMIC-III data set is publicly available (`https://github.com/DPBayes/individual-accounting-gdp`), as requested by Physionet (`https://physionet.org/content/mimiciii/view-dua/1.4/`).

This work was supported by the Academy of Finland (Flagship programme: Finnish Center for Artificial Intelligence, FCAI; and grant 325573), the Strategic Research Council at the Academy of Finland (Grant 336032) as well as the European Union (Project 101070617). Views and opinions expressed are however those of the author(s) only and do not necessarily reflect those of the European Union or the European Commission. Neither the European Union nor the granting authority can be held responsible for them.

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

# A   Existing Analysis Using RDP by Feldman and Zrnic (2021)

We next illustrate how the stochastic process $X_n$ that is used to analyse fully adaptive compositions is determined in case of RDP analysis (Feldman and Zrnic, 2021). Central in the analysis is showing the supermartingale property (A.5) of $X_n$.

The Rényi differential privacy (RDP) (Mironov, 2017) is defined as follows. Rényi divergence of order $\alpha \in (1, \infty)$ between two distributions $P$ and $Q$ is defined as

$$D_\alpha(P||Q) = \frac{1}{\alpha - 1} \log \int \left( \frac{P(t)}{Q(t)} \right)^\alpha Q(t) \, dt.$$

By continuity, for $\alpha = 1$ we have that $\lim_{\alpha \to 1+} D_\alpha(P||Q)$ equals the KL divergence $KL(P||Q)$.

**Definition A.1.** *We say that a mechanism $\mathcal{M}$ is $(\alpha, \rho)$-RDP, if for all neighbouring datasets $X, X'$, the output distributions $\mathcal{M}(X)$ and $\mathcal{M}(X')$ have Rényi divergence of order $\alpha$ less than $\rho$, i.e.*

$$\max_{X \simeq X'} \{ D_\alpha\big(\mathcal{M}(X)||\mathcal{M}(X')\big), D_\alpha\big(\mathcal{M}(X')||\mathcal{M}(X)\big) \} \leq \rho.$$

The fully adaptive RDP analysis by Feldman and Zrnic (2021) is based on studying the properties of the supermartingale $M_n$ which they define as

$$M_n(X, X') = \text{Loss}(a^{(n)}, X, X', \alpha) \cdot e^{-(\alpha-1) \sum_{m=1}^n \rho_m},$$

where $\alpha \geq 1$,

$$\text{Loss}(a^{(n)}, X, X', \alpha) = \left( \frac{\mathbb{P}(\mathcal{M}^{(n)}(X) = a^{(n)})}{\mathbb{P}(\mathcal{M}^{(n)}(X') = a^{(n)})} \right)^\alpha,$$

and $\rho_m$ gives the RDP of order $\alpha$ given $\mathcal{M}_{1:m-1}(X)$, i.e.

$$\rho_m = \frac{1}{\alpha - 1} \log \sup_{(X,X') \in \mathcal{S}} \mathbb{E}_{a^{(m)} \sim \mathcal{M}^{(m)}(X')} \left[ \left( \frac{\mathbb{P}(\mathcal{M}^{(m)}(X) = a^{(m)})}{\mathbb{P}(\mathcal{M}^{(m)}(X') = a^{(m)})} \right)^\alpha \bigg| a^{(m-1)} \right], \tag{A.1}$$

where $\mathcal{S}$ is a pre-determined set of neighbouring datasets In particular, the RDP bounds for the fully adaptive compositions are obtained by showing that $M_n(X, X')$ has the supermartingale property, meaning that

$$\mathbb{E}(M_n(X, X')|\mathcal{F}_{n-1}) \leq M_{n-1}(X, X'). \tag{A.2}$$

Feldman and Zrnic (2021) show that from this property, and from the law of total expectation (A.4), it follows that if $\sum_{i=1}^k \rho_i \leq B$ almost surely, where $K$ is the maximum number of compositions, then the fully adaptive composition is $(\alpha, B)$-RDP (Thm. 3.1, Feldman and Zrnic, 2021) .

Due to the factorisability of the Rényi divergence, the property (A.2) is straightforward to show for the random variable $M_n(X, X')$ using the Bayes theorem:

$$\begin{aligned}
&\mathbb{E}(M_n(X, X')|\mathcal{F}_{n-1}) \\
&= \mathbb{E}(\text{Loss}(a^{(n)}, X, X', \alpha) \cdot e^{-(\alpha-1) \sum_{m=1}^n \rho_m} |\mathcal{F}_{n-1}) \\
&= \mathbb{E} \left( \frac{\mathbb{P}(\mathcal{M}_n(X) = a_n|a^{(n-1)})}{\mathbb{P}(\mathcal{M}_n(X') = a_n|a^{(n-1)})} \right)^\alpha \cdot \text{Loss}(a^{(n-1)}, X, X', \alpha) \cdot e^{-(\alpha-1) \sum_{m=1}^n \rho_m},
\end{aligned} \tag{A.3}$$

since $\rho_1, \ldots, \rho_n \in \mathcal{F}_{n-1}$ and $\text{Loss}(a^{(n-1)}, X, X', \alpha) \in \mathcal{F}_{n-1}$. Moreover, as $\mathcal{M}_n$ is $\rho_n$-RDP,

$$\text{Loss}(a^{(n-1)}, X, X', \alpha) \leq e^{(\alpha-1)\rho_n},$$

and the supermartingale property follows from (A.3), i.e., that

$$\mathbb{E}(M_n(X, X')|\mathcal{F}_{n-1}) \leq M_{n-1}(X, X').$$

As the hockey-stick divergence does not factorise in this way, we need to take another approach to get the required supermartingale.

## A.1  Informal Description: Filtrations, Supermartingales, Stopping Times

Similarly to (Whitehouse et al., 2022) and (Feldman and Zrnic, 2021), we use the notions of filtrations and supermartingales for analysing fully adaptive compositions, where the individual worst-case pairs of distributions are not fixed but can be chosen adaptively based on the outcomes of the previous mechanisms. Given a probability space $(\Omega, \mathcal{F}, \mathbb{P})$, a filtration $(\mathcal{F}_n)_{n \in \mathbb{N}}$ of $\mathcal{F}$ is a sequence of $\sigma$-algebras satisfying: (i) $\mathcal{F}_n \subset \mathcal{F}_{n+1}$ for all $n \in \mathbb{N}$, and (ii) $\mathcal{F}_n \subset \mathcal{F}$ for all $n \in \mathbb{N}$. In the context of the so called natural filtration generated by a stochastic process $X_t$, $t \in \mathbb{N}$, the $\sigma$-algebra of the filtration $\mathcal{F}_n$ represents all the information contained in the first $n$ random variables $(X_1, \ldots, X_n)$. I.e., given $\mathcal{F}_n$, we know all possible events/outcomes that could have occurred up to the time-step $n$. The law of total expectation states that if a random variable $X$ is $\mathcal{F}_n$-measurable and $\mathcal{F}_n \subset \mathcal{F}_{n+1}$, then $\mathbb{E}((X|\mathcal{F}_{n+1})|\mathcal{F}_n) = \mathbb{E}(X|\mathcal{F}_n)$. Thus, if we have natural filtrations $\mathcal{F}_0, \ldots, \mathcal{F}_n$ for a stochastic process $X_0, \ldots, X_n$, then

$$\mathbb{E}(X_n|\mathcal{F}_0) = \mathbb{E}(((X_n|\mathcal{F}_{n-1})|\mathcal{F}_{n-2})|\ldots|\mathcal{F}_0). \tag{A.4}$$

The supermartingale property means that for all $n$,

$$\mathbb{E}(X_n|\mathcal{F}_{n-1}) \leq X_{n-1}. \tag{A.5}$$

From the law of total expectation it then follows that for all $n \in \mathbb{N}$,

$$\mathbb{E}(X_n|\mathcal{F}_0) \leq X_0.$$

We follow the analysis of Feldman and Zrnic (2021) and first set a maximum number of steps (denote by $k$) for the compositions. We do not release more information if a pre-defined privacy budget is exceeded. Informally speaking, the stochastic process $X_n$ that we analyse represents the sum of the realised privacy loss up to step $n$ and the budget remaining at that point. The privacy budget has to be constructed such that the amount of the budget left at step $n$ is included in the filtration $\mathcal{F}_{n-1}$. This allows us to reduce the privacy loss of the adaptively chosen $n$th mechanism from the remaining budget. Mathematically, this means that the integration $\mathbb{E}(X_n|\mathcal{F}_{n-1})$ will be only w.r.t. the outputs of the $n$th mechanism. Consider e.g. the differentially private version of the gradient descend (GD) method, where the amount of increase in the privacy budget depends on the gradient norms which depend on the parameter values at step $n-1$, i.e., they are included in $\mathcal{F}_{n-1}$. Then, $\mathbb{E}(X_k|\mathcal{F}_0)$ corresponds to the total privacy loss. If we can show that (A.5) holds for $X_n$, then by the law of total expectation the total privacy loss is less than $X_0$, the pre-defined budget. In our case the total budget $X_0$ will equal the $(\varepsilon, \delta)$-curve for a $\mu$-Gaussian DP mechanism, where $\mu$ determines the total privacy budget, and $\mathbb{E}[X_T]$, the expectation of the consumed privacy loss at step $T$, will equal the $(\varepsilon, \delta)$-curve for the fully adaptive composition to be analysed.

A discrete-valued stopping time $\tau$ is a random variable in the probability space $(\Omega, \mathcal{F}, \{\mathcal{F}_t\}, \mathbb{P})$ with values in $\mathbb{N}$ which gives a decision of when to stop. It must be based only on the information present at time $t$, not on any future information, i.e., it has to hold $\{\tau = t\} \in \mathcal{F}_t$. The optimal stopping theorem states that if the stochastic process $X_n$ is a supermartingale and if $T$ is a stopping time, then $\mathbb{E}[X_T] \leq \mathbb{E}[X_0]$. In the analysis of fully adaptive compositions, the stopping time $T$ will equal the step where the privacy budget is about to exceed the limit $B$. Then, only the outputs of the (adaptively selected) mechanisms up to step $T$ are released, and from the optimal stopping theorem it follows that $\mathbb{E}[X_T] \leq X_0$.

## B  Main Theorem

**Theorem B.1.** *Let $k$ denote the maximum number of compositions. Suppose that, almost surely,*

$$\sum_{m=1}^{k} \mu_m^2 \leq B^2. \tag{B.1}$$

*Then, $\mathcal{M}^{(k)}(X)$ is $B$-GDP.*

*Proof.* First, recall the notation from Section A: $\mathcal{L}_{X/X'}^{(k)}$ denotes the privacy loss between $\mathcal{M}^{(k)}(X)$ and $\mathcal{M}^{(k)}(X')$ with outputs $a^{(k)}$. Let $\varepsilon \in \mathbb{R}$. Our proof is based on showing the supermartingale

property for the random variable $M_n(\varepsilon)$, $n \in [k]$, defined as

$$M_k(\varepsilon) = \left[ 1 - e^{\varepsilon - \mathcal{L}_{X/X'}^{(k)}} \right]_+ ,$$

$$M_n(\varepsilon) = \mathop{\mathbb{E}}_{t \sim R_n} \left[ 1 - e^{\varepsilon - \mathcal{L}_{X/X'}^{(n)} - \mathcal{L}_n(t)} \right]_+ , \quad 0 \le n \le k-1, \qquad \text{(B.2)}$$

where $\mathcal{L}_n(t) = \log R_n(t)/Q(t)$ and $R_n$ is the density function of $\mathcal{N}\left( \sqrt{B^2 - \sum_{m=1}^n \mu_m^2}, 1 \right)$ and $Q$ is the density function of $\mathcal{N}(0,1)$. Moreover,

$$M_0(\varepsilon) = \mathop{\mathbb{E}}_{t \sim R_0} \left[ 1 - e^{\varepsilon - \mathcal{L}_0(t)} \right]_+ ,$$

where $\mathcal{L}_0(t) = \log R_0(t)/Q(t)$ and $R_0$ is the density function of $\mathcal{N}(B, 1)$ and $Q$ is the density function of $\mathcal{N}(0,1)$. Notice that in particular this means that $M_0(\varepsilon)$ gives $\delta(\varepsilon)$ for a $B$-GDP mechanism.

We next show that $\mathbb{E}\left[ M_k(\varepsilon) \middle| \mathcal{F}_{k-1} \right] \le M_{k-1}(\varepsilon)$.

Since the pair of distributions $\left( \mathcal{N}(\mu_k, 1), \mathcal{N}(0,1) \right)$ dominates the mechanism $\mathcal{M}_k$, we have by the Bayes rule and Corollary 7 that

$$\mathbb{E}\left[ M_k(\varepsilon) \middle| \mathcal{F}_{k-1} \right] = \mathop{\mathbb{E}}_{a^{(k)} \sim \mathcal{M}^{(k)}} \left[ \left[ 1 - e^{\varepsilon - \mathcal{L}_{X/X'}^{(k)}} \right]_+ \middle| \mathcal{F}_{k-1} \right]$$

$$= \mathop{\mathbb{E}}_{a^k \sim \mathcal{M}_k} \left[ \left[ 1 - e^{\varepsilon - \mathcal{L}_{X/X'}^{(k-1)} - \mathcal{L}_{X/X'}^{k}} \right]_+ \middle| \mathcal{F}_{k-1} \right]$$

$$\le \mathop{\mathbb{E}}_{t \sim P_k} \left[ 1 - e^{\varepsilon - \mathcal{L}_{X/X'}^{(k-1)} - \widetilde{\mathcal{L}}_k(t)} \right]_+ ,$$

where $\widetilde{\mathcal{L}}_k(t) = \log P_k(t)/Q(t)$ and $P_k$ is the density function of $\mathcal{N}(\mu_k, 1)$ and $Q$ is the density function of $\mathcal{N}(0,1)$. Above we have also used the fact that $\mathcal{L}_{X/X'}^{(k-1)} \in \mathcal{F}_{k-1}$.

Since $\sum_{m=1}^k \mu_m^2 \le B^2$ almost surely, i.e., $\mu_k \le \sqrt{B^2 - \sum_{m=1}^{k-1} \mu_m^2}$ almost surely, by the data-processing inequality for $\alpha$-divergence we have that, almost surely,

$$\mathop{\mathbb{E}}_{t \sim P_k} \left[ 1 - e^{\varepsilon - \mathcal{L}_{X/X'}^{(k-1)} - \widetilde{\mathcal{L}}_k(t)} \right]_+ \le \mathop{\mathbb{E}}_{t \sim R_{k-1}} \left[ 1 - e^{\varepsilon - \mathcal{L}_{X/X'}^{(k-1)} - \mathcal{L}_{k-1}(t)} \right]_+ = M_{k-1}(\varepsilon),$$

where $\mathcal{L}_{k-1}(t) = \log R_{k-1}(t)/Q(t)$ and $R_{k-1}$ is the density function of $\mathcal{N}\left( \sqrt{B^2 - \sum_{m=1}^{k-1} \mu_m^2}, 1 \right)$ and $Q$ is the density function of $\mathcal{N}(0,1)$. Therefore, $\mathbb{E}\left[ M_k(\varepsilon) \middle| \mathcal{F}_{k-1} \right] \le M_{k-1}(\varepsilon)$.

We next show that $\mathbb{E}\left[ M_{k-1}(\varepsilon) \middle| \mathcal{F}_{k-2} \right] \le M_{k-2}(\varepsilon)$. The supermartingale property follows then by induction.

Since $\mu_1, \ldots, \mu_{k-1} \in \mathcal{F}_{k-2}$, we have that, almost surely,

$$
\mathbb{E}\left[M_{k-1}(\varepsilon)\Big|\mathcal{F}_{k-2}\right] = \mathop{\mathbb{E}}_{a^{(k-1)}\sim\mathcal{M}^{(k-1)}}\left[\mathop{\mathbb{E}}_{t\sim R_{k-1}}\left[1 - \mathrm{e}^{\varepsilon-\mathcal{L}_{X/X'}^{(k-1)}-\mathcal{L}_{k-1}(t)}\right]_+\Bigg|\mathcal{F}_{k-2}\right]
$$

$$
= \mathop{\mathbb{E}}_{a^{k-1}\sim\mathcal{M}_{k-1}}\left[\mathop{\mathbb{E}}_{t\sim R_{k-1}}\left[1 - \mathrm{e}^{\varepsilon-\mathcal{L}_{X/X'}^{(k-2)}-\mathcal{L}_{X/X'}^{k-1}-\mathcal{L}_{k-1}(t)}\right]_+\Bigg|\mathcal{F}_{k-2}\right]
$$

$$
= \mathop{\mathbb{E}}_{t\sim R_{k-1}}\mathop{\mathbb{E}}_{a^{k-1}\sim\mathcal{M}_{k-1}}\left[1 - \mathrm{e}^{\varepsilon-\mathcal{L}_{X/X'}^{(k-2)}-\mathcal{L}_{X/X'}^{k-1}-\mathcal{L}_{k-1}(t)}\right]_+ \tag{B.3}
$$

$$
\leq \mathop{\mathbb{E}}_{t\sim R_{k-1}}\mathop{\mathbb{E}}_{t_{k-1}\sim P_{k-1}}\left[1 - \mathrm{e}^{\varepsilon-\mathcal{L}_{X/X'}^{(k-2)}-\widetilde{\mathcal{L}}_{k-1}(t_{k-1})-\mathcal{L}_{k-1}(t)}\right]_+
$$

$$
= \mathop{\mathbb{E}}_{t\sim R_{k-2}}\left[1 - \mathrm{e}^{\varepsilon-\mathcal{L}_{X/X'}^{(k-2)}-\mathcal{L}_{k-2}(t)}\right]_+
$$

$$
= M_{k-2},
$$

where $\widetilde{\mathcal{L}}_{k-1}(t) = \log P_{k-1}(t)/Q(t)$ and $P_{k-1}$ is the density function of $\mathcal{N}(\mu_{k-1}, 1)$ and $Q$ is the density function of $\mathcal{N}(0,1)$. In the inequality step we use Corollary 7 and the fact that the pair of distributions $(\mathcal{N}(\mu_{k-1}, 1), \mathcal{N}(0,1))$ dominates the mechanism $\mathcal{M}_{k-1}(\varepsilon)$. In the second last step we have also use the fact that if $\widehat{P}_1 \sim \mathcal{N}(\widehat{\mu}_1, 1)$, $\widehat{P}_1 \sim \mathcal{N}(\widehat{\mu}_2, 1)$ and $Q \sim \mathcal{N}(0,1)$, and $\widehat{\mathcal{L}}_1(t) = \log \widehat{P}_1(t)/Q(t)$ and $\widehat{\mathcal{L}}_2(t) = \log \widehat{P}_2(t)/Q(t)$, then

$$
\mathop{\mathbb{E}}_{t_1\sim\widehat{P}_1}\mathop{\mathbb{E}}_{t_2\sim\widehat{P}_2}\left[1 - \mathrm{e}^{\varepsilon-\widehat{\mathcal{L}}_1(t)-\widehat{\mathcal{L}}_2(t)}\right]_+ = \mathop{\mathbb{E}}_{t\sim\widehat{P}_3}\left[1 - \mathrm{e}^{\varepsilon-\widehat{\mathcal{L}}_3(t)}\right]_+,
$$

where $\widehat{P}_3 \sim \mathcal{N}(\sqrt{\widehat{\mu}_1^2 + \widehat{\mu}_2^2}, 1)$ and $\widehat{\mathcal{L}}_3(t) = \log \widehat{P}_3(t)/Q(t)$. This follows directly from the fact that the PLDs determined by the pairs of distributions $(\widehat{P}_1, Q)$ and $(\widehat{P}_2, Q)$ are Gaussians (see Eq. (2.2)), the convolution of two Gaussians is a Gaussian.

By induction, we see from (B.3) that the the supermartingale property holds for the random variable $M_n(\varepsilon)$. By the law of total expectation (A.4), $\mathbb{E}[M_k(\varepsilon)] \leq M_0(\varepsilon)$. By Theorem 6,

$$
\mathbb{E}[M_k(\varepsilon)] = H_{\mathrm{e}^\varepsilon}\big(\mathcal{M}^{(k)}(X)||\mathcal{M}^{(k)}(X')\big),
$$

and

$$
M_0(\varepsilon) = H_{\mathrm{e}^\varepsilon}\big(\mathcal{N}(B, 1)||\mathcal{N}(0,1)\big).
$$

As $\varepsilon$ was taken to be an arbitrary real number, the inequality $\mathbb{E}[M_k(\varepsilon)] \leq M_0(\varepsilon)$ holds for all $\varepsilon \in \mathbb{R}$ and by Lemma 9 we see that $\mathcal{M}^{(k)}(X)$ is $B$-GDP. $\qquad\square$

## C Filters and Odometers

We here give additional details on the GDP filters and shortly discuss implementation of GDP privacy odometers.

### C.1 GDP - privacy filter

For simplicity, we here consider a GDP filter that chooses the privacy parameters adaptively, but not individually (like the filter in the main text). I.e., the amount that the privacy budget is spent at each step has to provide a guarantee over the whole data set.

To this end we formally define a GDP filter as

$$
\mathcal{F}_B(\mu_1, \ldots, \mu_t) = \begin{cases} \mathrm{HALT}, & \text{if } \sum_{i=1}^t \mu_i > B, \\ \mathrm{CONT}, & \text{else.} \end{cases}
$$

Using the filter $\mathcal{F}_B$, a GDP filter is given as in Alg. 2.

In principle, the supermartingale property of the random variable $M_n(\varepsilon)$, as defined in (B.2), s sufficient to show that the algorithm below is $B$-GDP. The only difference is that the algorithm can stop at random time. To include that feature in the analysis, we need to use the optimal stopping time theorem.

---

**Algorithm 2** GDP Filter Algorithm

---

Input: Budget $B$, maximum number of compositions $k$, initial value $a_0$.
**for** $j = 1, \ldots, k$ **do**
    Find parameter $\mu_j \geq 0$ such that $\mathcal{M}_j(\cdot, a^{(j-1)})$ is $\mu_j$-GDP.
    **if** $\sum_{\ell=0}^{j} \mu_\ell^2 > B^2$: **then**
       BREAK
    **else**
       $a_j = \mathcal{M}_j(X, a^{(j-1)})$
    **end if**
**end for**
**return** $a^{(j-1)}$.

---

**Theorem C.1.** *Denote by $\mathcal{M}$ the output of Algorithm 2. $\mathcal{M}$ is $B$-GDP under remove neighbourhood relation, meaning that for all datasets $X \in \mathcal{X}^N$, for all $i \in [N]$ and for all $\alpha > 0$:*

$$\max\{H_\alpha\big(\mathcal{M}(X)\|\mathcal{M}(X^{-i}))\big), H_\alpha\big(\mathcal{M}(X^{-i})\|\mathcal{M}(X))\big)\} \leq H_\alpha\big(\mathcal{N}(B,1)\|\mathcal{N}(0,1)\big). \quad \text{(C.1)}$$

*Proof.* The proof goes exactly the same as the proof for (Thm. 4.3 Feldman and Zrnic, 2021) which holds for the RDP filter. By using the fact that for all $t \geq 0$: $\mu_{t+1} \in \mathcal{F}_t$, where $\mathcal{F}_t$ is the natural filter $\sigma(a^{(t)})$, we see that the random variable

$$T = \min\{t \,:\, \mathcal{F}_B(\mu_1, \ldots, \mu_{t+1}) = \text{HALT}\} \wedge k$$

is a stopping time since $\{T = t\} \in \mathcal{F}_t$ since $\mu_{t+1} \in \mathcal{F}_t$. From the optimal stopping theorem and the supermartingale property it then follows that for all $\varepsilon \in \mathbb{R}$, $\mathbb{E}[M_T(\varepsilon)] \leq M_0(\varepsilon)$, which by the reasoning of the proof of Thm.10 shows that (C.1) holds, i.e., output of Alg. 2 is $B$-GDP w.r.t. to the removal neighbourhood relation of data sets. $\square$

A benefit of GDP filter when compared to RDP filter is that we obtain tight $(\varepsilon, \delta(\varepsilon))$-bounds for adaptive compositions of Gaussian mechanisms. Moreover, from Thm. 9 it follows that these tight $(\varepsilon, \delta(\varepsilon))$-DP bounds can be obtained by an analytic formula:

**Corollary C.2.** *The output of Algorithm 2 is $(\varepsilon, \delta(\varepsilon))$-DP for all $\varepsilon \geq 0$ and*

$$\delta(\varepsilon) = \Phi\left(-\frac{\varepsilon}{B} + \frac{B}{2}\right) - e^\varepsilon \, \Phi\left(-\frac{\varepsilon}{B} - \frac{B}{2}\right). \quad \text{(C.2)}$$

### C.2 Tight Bounds for the Gaussian Mechanism

When running e.g. the DP-GD algorithm and using either the filtering of Alg. 2 or the individual filtering of Alg. 1, by appropriate scaling of the gradients each individual data element can be made to fully consume its privacy budget. This scaling for individual filtering is given in (Algorithm 3 Feldman and Zrnic, 2021).

**Remark C.3.** *Suppose we use the Gaussian mechanism and scale the noise for each data element $x_i$, $i \in [N]$ at the last step such that the GDP budget is fully consumed, i.e., we have that $\sum_j \mu_j^{(i)} = B$, then the resulting algorithm is tightly $(\varepsilon, \delta)$-DP for $\delta(\varepsilon)$ given by the expression* (C.2), *in a sense that for all $i \in [N]$,*

$$\max\{H_{e^\varepsilon}\big(\mathcal{M}^{(k)}(X)\|\mathcal{M}^{(k)}(X^{-i}))\big), H_\alpha\big(\mathcal{M}^{(k)}(X^{-i})\|\mathcal{M}^{(k)}(X))\big)\} = \delta(\varepsilon).$$

### C.3 Benefits of GDP vs. RDP Filtering

To experimentally illustrate the benefits of GDP accounting, consider one of the private GD experiments of (Feldman and Zrnic, 2021), where $\sigma = 100$, and number of compositions corresponding to worst-case analysis is $k = 420$. The RDP value of order $\alpha$ corresponding to this iteration is then $\alpha/(2 \cdot \widetilde{\sigma}^2)$, where $\widetilde{\sigma} = \sigma/\sqrt{k}$. Figure 3 shows the $(\varepsilon, \delta)$-values, computed via RDP and GDP. To get the $(\varepsilon, \delta)$-values from the RDP-values, we use the conversion included in the TensorFlow Privacy library. When using GDP instead of RDP, we can run $k = 500$ iterations instead of the $k = 420$ iterations, for an equal privacy budget.

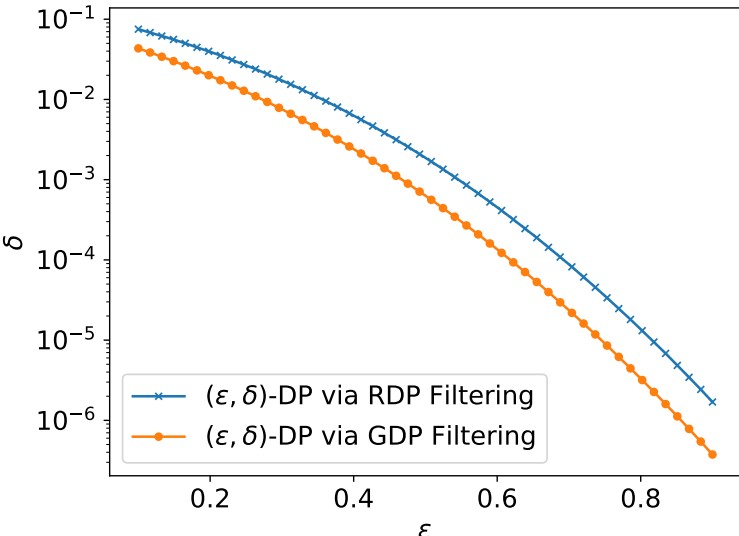

Figure 3: Comparison of RDP and GDP for private GD with $\sigma = 100$ and number of iterations $k = 420$. This means that when we replace the RDP filter with GDP filter for the private GD, we can get roughly 10 percent smaller $\varepsilon$-values.

### C.4 GDP Privacy Odometers

We here also shortly comment on privacy odometers, considered e.g. by (Rogers et al., 2016; Feldman and Zrnic, 2021; Lécuyer, 2021).

In practice, one might want to track the privacy loss incurred so far. Rogers et al. (2016) were the first ones to formalise this in terms of a privacy odometer. Feldman and Zrnic (2021) utilise a sequence of valid Rényi privacy filters such that a fixed sequence of privacy losses $B_1, B_2, \ldots$ determine random stopping times $T_1, T_2, \ldots$ such that the privacy spent up to time $T_i$ is at most $B_i$. By assuming, for example, that for all $i$, $B_{i+1} - B_i = \Delta$ for a fixed discretisation parameter $\Delta > 0$, we may employ the RDP filter such that whenever the privacy budget counter crosses $\Delta$ (suppose for the $m$th time) we release the sequence $a^{(T_m)}$ and initialize the privacy loss counter to zero. The fact that $a^{(T_m)}$ is $m\Delta$-RDP follows directly from the RDP results that hold for the filters.

With the GDP, we can construct in the exactly same way an algorithm that outputs states always after every predetermined amount of GDP budget $\Delta$ is spent. If at round $i$ we spend $\Delta_i$-GDP budget, by the results for GDP filters we know that $B_{i+1}^2 = B_i^2 + \Delta_i^2$, and that the output $a^{(T_m)}$ is $\widetilde{B}_m$-GDP, where $\widetilde{B}_m = \sqrt{\Delta_1^2 + \ldots + \Delta_m^2}$.

## D    Approximative $(\varepsilon, \delta)$-Filter via Blackwell's Theorem

We next consider a filter that can use any individual dominating pairs of distributions, not just Gaussians. To this end, we need find determine pairs of dominating distributions at each iteration.

**Assumption.** Given neighbouring data sets $X, X'$, we assume that for all $n$, we can determine a dominating pair of distributions $(P_n, Q_n)$ such that for all $\alpha > 0$,

$$H_\alpha\big(\mathcal{M}_n(a^{(n-1)}, X)||\mathcal{M}_n(a^{(n-1)}, X')\big) \leq H_\alpha\big(P_n||Q_n\big).$$

A tightly dominating pair of distributions $(P_n, Q_n)$ always exists (Proposition 8, Zhu et al., 2022), and on the other hand, uniquely determining such a pair is straightforward for the subsampled Gaussian mechanism, for example (see Lemma 8). For the so called shufflers, such worst case pairs can be obtained by post-processing (Feldman et al., 2022). From the Blackwell theorem (Dong et al., 2022, Thm. 2.10) it follows that there exists a stochastic transformation (Markov process) $T$ such that

$T\mathcal{M}_n(a^{(n-1)}, X) = P_n$ and $T\mathcal{M}_n(a^{(n-1)}, X') = Q_n$. By the data-processing inequality for Rényi divergence the pair $(P_n, Q_n)$ is then similarly a worst-case pair for RDP, for all orders $\alpha \geq 1$.

First, we replace the GDP filter condition $\sum \mu_i^2 \leq B^2$ by the condition ($\mu > 0$)

$$\mathbb{E}_{t_1 \sim P_1, \ldots, t_n \sim P_n} \left[ 1 - e^{\varepsilon - \sum_{m=1}^n \mathcal{L}_m(t_m)} \right]_+ \leq H_{e^\varepsilon}\big(\mathcal{N}(\mu, 1) || \mathcal{N}(0, 1)\big) \tag{D.1}$$

for all $\varepsilon \in \mathbb{R}$. By the Blackwell theorem there exists a stochastic transformation that maps the product distributions $(P_1, \ldots, P_n)$ and $(Q_1, \ldots, Q_n)$ to distributions for which the hockey-stick divergence is $H_{e^\varepsilon}\big(\mathcal{N}(0, 1) || \mathcal{N}(\mu, 1)\big)$ for all $\varepsilon \in \mathbb{R}$. By the data-processing inequality we have

$$D_\alpha(P_1 \times \ldots \times P_n || Q_1 \times \ldots \times Q_n) \leq D_\alpha\big(\mathcal{N}(0, 1) || \mathcal{N}(\mu, 1)\big), \tag{D.2}$$

for all $\alpha \geq 1$, where $D_\alpha$ denotes the Rényi divergence of order $\alpha$. Since the pairs $(P_i, Q_i)$, $1 \leq i \leq n$, are the worst-case pairs also for RDP (as described above), by (D.2) and the RDP filter results of Feldman and Zrnic (2021), we have that for all $\alpha \geq 1$,

$$D_\alpha\big(\mathcal{M}(X) || \mathcal{M}(X')\big) \leq D_\alpha\big(\mathcal{N}(\mu, 1) || \mathcal{N}(0, 1)\big). \tag{D.3}$$

By converting the RDPs of Gaussians in (D.3) to $(\varepsilon, \delta)$-bounds, this procedure provides $(\varepsilon, \delta)$-upper bounds and can be straightforwardly modified into an individual PLD filter as in case of GDP. One difficulty, however, is how to compute the parameter $\mu$ in (D.1), given the individual pairs $(P_i, Q_i)$, $1 \leq i \leq n$. When the number of iterations is large, by the central limit theorem the PLD of the composition starts to resemble that of a Gaussian mechanism (Sommer et al., 2019), and it is then easy to numerically approximate $\mu$ (see Fig. 1 for an example). It is well known that the $(\varepsilon, \delta)$-bounds obtained via RDPs are always non-tight, since the conversion of RDP to $(\varepsilon, \delta)$ is lossy (Zhu et al., 2022). Moreover, often the computation of the RDP values themselves is lossy. In the procedure described here, the only loss comes from converting (D.3) to $(\varepsilon, \delta)$-bounds. In Appendix E we show how to numerically efficiently compute the individual PLDs using FFT.

# E  Efficient Individual Numerical Accounting for DP-SGD

We next show how to compute the individual PLDs for DP-SGD. These are needed when implementing the approximative individual $(\varepsilon, \delta)$-accountant described in Section D. The errors arising from the approximations are generally negligible and a rigorous error analysis could be carried out using the techniques presented in (Koskela et al., 2021) and (Gopi et al., 2021).

The numerical approximation is based on

1. a numerical $\sigma$-grid which allows evaluating upper bounds for $\delta$'s efficiently: we precompute FFTs for different $\sigma$-values and no additional FFT computations are then needed during the evaluation of the individual $\varepsilon$-values. By the data-processing inequality this grid approximation also leads to upper $(\varepsilon, \delta)$-bounds.

2. the Plancherel Theorem, which removes the need to compute inverse FFTs when evaluating individual PLDs.

First, we recall some basics about numerical accounting using FFT (see also (Koskela et al., 2020; Gopi et al., 2021)).

## E.1  Numerical Evaluation of DP Parameters Using FFT

We use a Fast Fourier Transform (FFT)-based method by Koskela et al. (2020, 2021) called the Fourier Accountant (FA). The same approximation could be done when using the PRV accountant by Gopi et al. (2021). Using FFT requires that we truncate and place the PLD $\omega$ on an equidistant numerical grid over an interval $[-L, L]$, $L > 0$. Convolutions are evaluated using the FFT algorithm and using the error analysis the error incurred by the numerical FFT approximation can be bounded.

The Fast Fourier Transform (FFT) is described as follows (Cooley and Tukey, 1965). Let $x = [x_0, \ldots, x_{n-1}]^T$, $w = [w_0, \ldots, w_{n-1}]^T \in \mathbb{R}^n$. The discrete Fourier transform $\mathcal{F}$ and its inverse $\mathcal{F}^{-1}$ are defined as (Stoer and Bulirsch, 2013)

$$(\mathcal{F}x)_k = \sum_{j=0}^{n-1} x_j e^{-i\,2\pi kj/n}, \quad (\mathcal{F}^{-1}w)_k = \frac{1}{n} \sum_{j=0}^{n-1} w_j e^{i\,2\pi kj/n},$$

where i $= \sqrt{-1}$. Using FFT the running time of evaluating $\mathcal{F}x$ and $\mathcal{F}^{-1}w$ reduces to $O(n \log n)$. Also, FFT enables evaluating discrete convolutions efficiently using the so called convolution theorem

For obtaining computational speed-ups, we use the Plancherel Theorem (Chpt. 12, Press et al., 2007), which states that the DFT preserves inner products: for $x, y \in \mathbb{R}^n$,

$$\langle x, y \rangle = \frac{1}{n} \langle \mathcal{F}(x), \mathcal{F}(y) \rangle$$

When using FA to approximate $\delta(\varepsilon)$, we need to evaluate an expression of the form

$$\boldsymbol{b}^k = D\, \mathcal{F}^{-1}\big(\mathcal{F}(D\boldsymbol{a}^1)^{\odot k_1} \odot \cdots \odot \mathcal{F}(D\boldsymbol{a}^m)^{\odot k_m}\big), \quad D = \begin{bmatrix} 0 & I_{n/2} \\ I_{n/2} & 0 \end{bmatrix} \in \mathbb{R}^{n \times n},$$

where $\boldsymbol{a}^i$ corresponds to a numerical PLD for a combination of DP hyperparameters $i$, and $k_i$ is the number of times the composition contains a mechanism with this PLD.

Approximation for $\delta(\varepsilon)$ is then obtained from the discrete sum that approximates the hockey-stick integral:

$$\widetilde{\delta}(\varepsilon) = \sum_{-L+\ell\Delta x > \varepsilon} \big(1 - \mathrm{e}^{\varepsilon - (-L+\ell\Delta x)}\big)\, b_\ell^k.$$

The Plancherel Theorem gives the following:

**Lemma E.1.** *Let $\widetilde{\delta}(\varepsilon)$ and $\boldsymbol{b}^k$ be defined as above.*

*Denote $\boldsymbol{w}_\varepsilon \in \mathbb{R}^n$ such that*

$$(\boldsymbol{w}_\varepsilon)_\ell = \Big[0, 1 - \mathrm{e}^{\varepsilon - (-L + \ell\Delta x)}\Big]. \tag{E.1}$$

*Then, we have that*

$$\widetilde{\delta}(\varepsilon) = \frac{1}{n} \langle \mathcal{F}(D\boldsymbol{w}_\varepsilon), \mathcal{F}(D\boldsymbol{a}^1)^{\odot k_1} \odot \cdots \odot \mathcal{F}(D\boldsymbol{a}^m)^{\odot k_m} \rangle.$$

*Proof.* See (Koskela and Honkela, 2021). □

We instantly see that if both $\mathcal{F}(D\boldsymbol{w}_\varepsilon)$ and $\mathcal{F}(D\boldsymbol{a}^i)$, $1 \le i \le m$, are precomputed, $\widetilde{\delta}(\varepsilon)$ can be computed in $\mathcal{O}(n)$ time (where $n$ is the number of discretisation points for the PLD).

We can utilise this by placing the individual DP hyperparameters into well-chosen buckets, and by pre-computing FFTs corresponding to the hyperparameter values of each bucket. Then, the approximative numerical PLD for each sequence of DP hyperparameters (e.g. sequence of noise ratios) can be written in a form

$$\mathcal{F}(D\boldsymbol{a}^1)^{\odot k_1} \odot \cdots \odot \mathcal{F}(D\boldsymbol{a}^m)^{\odot k_m},$$

where $k_i$'s correspond to number of elements in each bucket. If we also have $\mathcal{F}(D\boldsymbol{w}_\varepsilon)$ precomputed for different values of $\varepsilon$, we can easily construct a numerical accountant that outputs an approximation of $\varepsilon$ as a function of $\delta$.

### E.2 Noise Variance Grid for Fast Individual Accounting

We next show how to construct the DP hyperparameter grid for DP-SGD: a numerical $\sigma$-grid. We remark that Yu et al. (2022) carry out similar approximations for speeding up their approximative individual RDP accountants.

Suppose we have models $a_0, \ldots, a_T$ as an output of DP-SGD iteration that we run with subsampling ratio $q$, clipping constant $C > 0$ and noise parameter $\sigma$. Also, suppose, that for a given data element $x$, along the iteration the clipped gradients have norms

$$C_{x,i} := \begin{cases} \|\nabla_\theta f(a_i, x)\|, & \text{if} \quad \|\nabla_\theta f(a_i, x)\| \le C, \\ C, & \text{else.} \end{cases}, \quad 0 \le i \le T - 1.$$

We get the individual $\varepsilon_x$-value (or individual numerical PLD, more generally) then for the entry $x$ by considering heterogeneous compositions of DP-SGD mechanisms with parameter values

$$q \quad \text{and} \quad \widetilde{\sigma}_{x,i} = \frac{C}{C_{x,i}} \cdot \sigma, \quad 0 \leq i \leq T - 1.$$

A naive approach would require up to $T$ different FFT evaluations which easily becomes computationally heavy. For the approximation, we determine a $\sigma$-grid

$$\Sigma = \{\sigma_0, \dots, \sigma_{n_\sigma}\},$$

where $n_\sigma \in \mathbb{Z}^+$ is a number of intervals in the grid and

$$\sigma_i = \sigma + i \cdot \frac{\sigma_{\max} - \sigma}{n_\sigma}.$$

We then encode the sequence of noise ratios

$$\widetilde{\Sigma} := \{\widetilde{\sigma}_{x,0}, \dots, \widetilde{\sigma}_{x,n-1}\}$$

into a tuple of integers

$$\mathbf{k} = (k_0, k_1, \dots, k_{n_\sigma}),$$

where

$$k_i = \begin{cases} \#\{\widetilde{\sigma} \in \widetilde{\Sigma} \ : \ \sigma_i \leq \widetilde{\sigma} < \sigma_{i+1}\}, & i < n_\sigma \\ \#\{\widetilde{\sigma} \in \widetilde{\Sigma} \ : \ \sigma_{n_\sigma} \leq \widetilde{\sigma}\}, & i = n_\sigma. \end{cases} \tag{E.2}$$

i.e. $k_i$ is number of scaled noise parameters $\widetilde{\sigma}$ hitting the bin number $i$ in the grid $\Sigma$.

By the construction of the approximation, we have the following:

**Theorem E.2.** *Consider the approximation described above. Denote the FFT transformation of the approximative numerical PLD obtained with the $\Sigma$-grid as*

$$\widetilde{a}_x = \mathcal{F}(D\widetilde{\boldsymbol{a}}^1)^{\odot k_1} \odot \cdots \odot \mathcal{F}(D\widetilde{\boldsymbol{a}}^{n_\sigma})^{\odot k_{n_\sigma}}$$

*and the corresponding $\delta$ (as a function of $\varepsilon$), as given by Lemma E.1 as*

$$\widetilde{\delta}(\varepsilon) = \frac{1}{n}\langle \mathcal{F}(D\boldsymbol{w}_\varepsilon), \widetilde{a}_x \rangle,$$

*where $\boldsymbol{w}_\varepsilon$ is the weight vector (E.1). Then, we have that for each $\varepsilon \geq 0$:*

$$\delta(\varepsilon) \leq \widetilde{\delta}(\varepsilon) + \mathrm{err},$$

*ehere $\delta(\varepsilon)$ is the tight value of $\delta$ corresponding to the actual sequence of noise ratios $\{\widetilde{\sigma}_{x,0}, \dots, \widetilde{\sigma}_{x,n-1}\}$ and $\mathrm{err}$ denotes the (controllable) numerical errors arising from the discretisation of PLDs.*

*Proof.* The results follows from the data-processing inequality since each $\sigma_{x,i}$-value is placed to bucket corresponding to a smaller noise ratio. $\qquad\square$

The numerical error term $\mathrm{err}$ can also be bounded using the techniques and results of (Gopi et al., 2021). The importang thing here is that if the FFTs $\mathcal{F}(D\widetilde{\boldsymbol{a}}^i)$, $0 \leq i \leq n_\sigma$, are precomputed as well as $\mathcal{F}(D\boldsymbol{w}_\varepsilon)$, then evaluating $\widetilde{\delta}(\varepsilon)$ is an $\mathcal{O}(n)$ operation.

To implement the approximative accountant described in Section D, we numerically approximate individual upper bound $\mu$-GDP values using the bisection method. The final individual $\varepsilon$-values are then obtained from the RDP to $(\varepsilon, \delta)$-conversion of (D.3) using the thus found individual $\mu$-values.

### E.3  Poisson Subsampling of the Gaussian Mechanism

For completeness we show how to determine the PLDs for the Poisson subsampled Gaussian mechanism, required for the individual accounting of DP-SGD. Consider the Gaussian mechanism

$$\mathcal{M}(X) = \sum_{x \in X} f(x) + \mathcal{N}(0, \sigma^2 I_d),$$

where $f$ is a function $f : \mathcal{X} \to \mathbb{R}^d$. Then, if the data set $X'$ and $X$ are neighbours such that $X = X' \bigcup \{x'\}$ for some entry $x'$, then from the translation invariance of the hockey-stick divergence and from the unitary invariance of the Gaussian noise, it follows that, for all $\alpha \geq 0$,

$$H_\alpha\big(\mathcal{M}_n(X)\|\mathcal{M}_n(X')\big) = H_\alpha\big(\mathcal{N}\big(\|f(x')\|_2, \sigma^2\big)\|\mathcal{N}\big(0, \sigma^2\big)\big).$$

Furthermore, from the scaling invariance of the hockey-stick divergence, we have that for all $\alpha \geq 0$,

$$H_\alpha\big(\mathcal{M}_n(X)\|\mathcal{M}_n(X')\big) = H_\alpha\big(\mathcal{N}\big(C, \sigma^2\big)\|\mathcal{N}\big(0, \sigma^2\big)\big)$$
$$= H_\alpha\big(\mathcal{N}\big(1, (\sigma/C)^2\big)\|\mathcal{N}\big(0, (\sigma/C)^2\big)\big),$$

where $C = \|f(x')\|_2$. Using the subsampling amplification results of Zhu et al. (2022) we get a unique worst-case pair $(P, Q)$, where

$$P = q \cdot \mathcal{N}\big(1, \widetilde{\sigma}^2\big) + (1 - q) \cdot \mathcal{N}\big(0, \widetilde{\sigma}^2\big),$$
$$Q = \mathcal{N}\big(0, \widetilde{\sigma}^2\big),$$

where $\widetilde{\sigma} = \sigma/C$. The PLD $\omega_{P/Q}$ is then determined by $P$ and $Q$ as defined in Def. 5.

## F  Experiments with MIMIC-III

We use the preprocessing provided by Harutyunyan et al. (2019) to obtain the train and test data for the phenotyping task. We refer to Harutyunyan et al. (2019) for details on the preprocessing pipeline and the details on the phenotyping task. We tune the hyperparameters using Bayesian optimization using the hyperparameter tuning library Optuna (Akiba et al., 2019) to maximise the macro-averaged AUC-ROC, the task's primary metric. We train using DP-GD and opacus (Yousefpour et al., 2021) with Poisson subsampling with noise parameter $\sigma \approx 10.61$ and determine the optimal clipping constant as $C \approx 0.79$. We compute the budget $B$ so that filtering starts after 50 epochs and set the maximum number of epochs to 100.

### F.1  Effect of Suboptimal Hyperparameter Values on Filtered DP-GD

We study also the effect of choosing sub-optimal clipping constants by evaluating the effects of filtering using clipping constants reaching from half the optimum to five times the optimum (refer to Figure 4). We observe that filtering only improves the utility when choosing clipping constants that are sub-optimal (e.g., 5x the optimum). Our observations complement the observations made by (Feldman and Zrnic, 2021), who also observe the largest improvements by filtering in sub-optimal hyperparamter regimes.

### F.2  Histograms of Individual $\varepsilon$-Values for the MIMIC-III Experiment

As described in the main text, to observe the differences across subgroups, we choose five non-overlapping groups of size 1000 based on the following criteria: subgroup 0: No diagnosis at all, subgroups 1 and 2: Pneumonia/no Pneumonia, subgroups 3 and 4: Heart attack/no heart attack. In the training data, there are total 2072 cases without a diagnosis, 4105 Pneumonia cases and in total 9413 heart attack cases. We remark that it is not uncommon for a patient to have multiple conditions.

During the training, we track the gradient norms $C_{x,i}$ for all elements of the training data set, and after the training, we compute the individual $\varepsilon$-values for $\delta = 10^{-5}$. In Figures 5 and 6 we display histograms of the individual $\varepsilon$ values after 50 epochs. With the optimal clipping constant a majority of the datapoints have a individual $\varepsilon = 2.75$, which is near the budget. For a clipping constant that is 5x the optimum most points are significantly smaller than $\varepsilon = 2.75$.

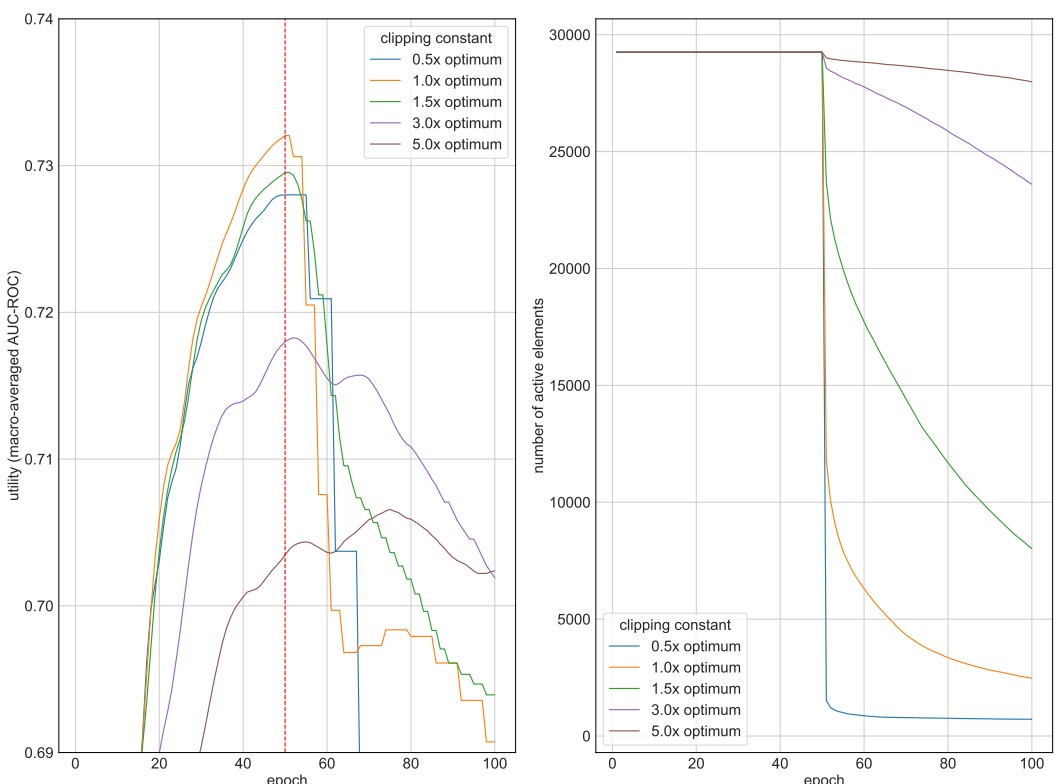

Figure 4: Filtered DP-GD with a maximum privacy loss of $(\varepsilon = 2.75, \delta = 1e\text{-}5)$ using tuned hyperparameters and different clipping constants. Left: The test AUC-ROC as a function of epochs. The red vertical line denotes the starting point of the filtering. Right: The number of active elements, which drops faster for a smaller clipping constant.

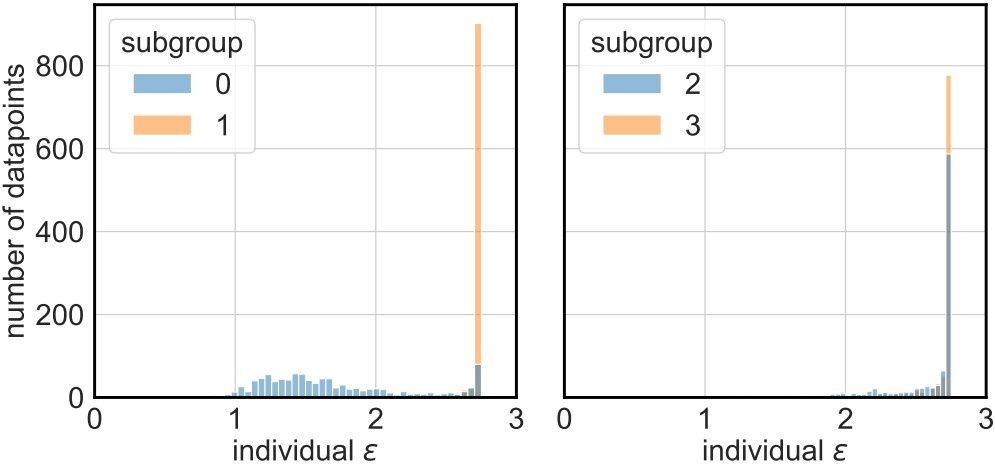

Figure 5: Histogram of individual $\varepsilon$ after 50 epochs without filtering when using the optimal clipping constant. A majority of the individual $\varepsilon$ are near $\varepsilon = 2.75$, which means that they will be deactivated in epoch 51, which uses filtering.

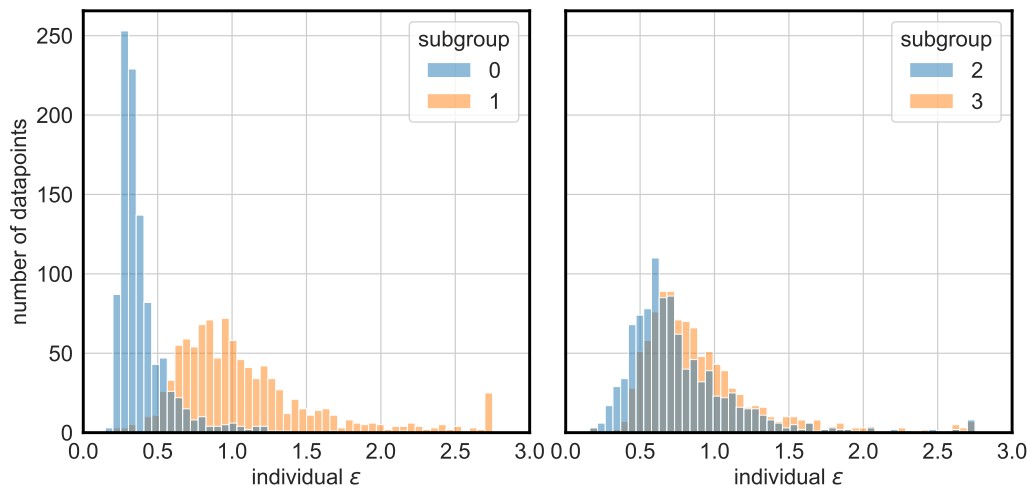

Figure 6: Histogram of individual $\varepsilon$ after 50 epochs without filtering when using a clipping constant that is 5x the optimum. A majority of the individual $\varepsilon$ are far away from $\varepsilon = 2.75$, which means that they will not be instantly deactivated in epoch 51, which uses filtering.

