# OpenReview forum: "Individual Privacy Accounting with Gaussian Differential Privacy"
_NeurIPS.cc/2022/Workshop/TSRML — TSRML2022_

### Official Review · Reviewer_sNuf · 2022-10-17

**Overall Rating:** 6

**Summary:**

This work proposes an extension of Feldman and Zrnic for individual privacy accounting using Gaussian Differential Privacy. This work shows by using GDP, one could obtain tighter individual privacy budgets compared to using RDP.

**Strengths:**

This work takes a step on extending existing result to GDP, which is a more advanced privacy accounting method. Empirically, this work also demonstrates the individual privacy accounting using GDP gives tighter \epsilon compared to RDP.

**Weaknesses:**

This work seems to add little novelty to the original Feldman and Zrnic work in terms of proof technique. The presentation also needs to be improved to highlight what is the main theoretical contribution (Thm 10 and Thm 11).

**Overall Recommendation:**

Since this work proposes an extension and improvement upon existing work, I recommend this work to be presented at the workshop.

**Review Confidence:**

3: The reviewer is fairly confident that the evaluation is correct

---

### Official Review · Reviewer_CfhD · 2022-10-21
**A clear and straightforward extension of individual privacy accounting for gaussian differential privacy.**

**Overall Rating:** 7

**Summary:**

The paper studies a fully adaptive individual DP analysis using the Gaussian differential privacy (GDP). Individual privacy accounting can be informative since the individual privacy losses are considerably smaller than those indicated by the DP bounds, thus resulting in tighter privacy analysis. The privacy loss computation is done for the fully adaptive composition setting, in which the input of the next mechanism is the output of the previous mechanism. The paper uses privacy filters to ensure that the current individual loss is less than a predefined privacy budget. To provide individual accounting analysis for GDP, the paper relies on determining a certain supermartingale for hockey-stick divergence and similar proof techniques as in the Renyi DP-based fully adaptive composition results of Feldman and Zrnic (2021). The paper claims that their technique in GDP leads to smaller $\epsilon$-values than commonly used RDP accountants. The proposed method is demonstrated in MINIST classification task and MIMIC-III multi-label classification task. The paper observes that individual filtering leads to a disparate loss of accuracies among subgroups during training.

**Strengths:**

- The presentation is clean and organized, and the proofs are quite clear and straight forward to follow. It’s nice to have a section that shows similar work for RDP in the appendix section so that reader can follow the general picture.
- The results on 2 different datasets show that their accounting techniques provide tighter bounds than other accountant techniques.

**Weaknesses:**

- Some minor typos in the proof: for example, in line 104, $\mathcal{M}^m \rightarrow \mathcal{M}^{(m)}$
- The paper isn’t too clear or doesn't have a good portion mentioning the individual privacy definition and the connection between the general definition of privacy to individual privacy. Readers that aren’t familiar with previous existing work for RDP may get confused.

**Overall Recommendation:**

This paper provides a nice extension of individual privacy accounting for GDP.  Individual privacy accounting is not novel but only work for RDP has been done. I recommend this paper.

**Review Confidence:**

3: The reviewer is fairly confident that the evaluation is correct

---

### Decision · Program_Chairs · 2022-10-23

**Decision:**

Accept

**Comment:**

Following the unanimous recommendations from reviewers, the submission is accepted.